# High Oxygen and Water-Vapor Transmission Rate and In Vitro Cytotoxicity Assessment with Illite-Polyethylene Packaging Films

**DOI:** 10.3390/ma13102382

**Published:** 2020-05-21

**Authors:** Dong Min Seong, Heysun Lee, Jungbae Kim, Jeong Ho Chang

**Affiliations:** 1Center for Convergence Bioceramic Materials, Korea Institute of Ceramic Engineering and Technology (KICET), Chungbuk 28160, Korea; sdm6230@naver.com (D.M.S.); hslee@kicet.re.kr (H.L.); 2Department of Chemical and Biological Engineering, Korea University, Seoul 02841, Korea; jbkim3@korea.ac.kr

**Keywords:** illite, polyethylene, composite film, cytotoxicity, oxygen transmission rate, water-vapor transmission rate

## Abstract

This work reports the preparation of a ceramic hybrid composite film with illite and polyethylene (illite-PE), and the evaluation of the freshness-maintaining properties such as oxygen transmission rate (OTR), water vapor transmission rate (WVTR), tensile strength, and in vitro cytotoxicity. The particle size of the illite material was controlled to within 10 μm. The illite-PE masterbatch and film were prepared using a twin-screw extruder and a blown film maker, respectively. The dispersity and contents of illite material in each masterbatch and composite film were analyzed using a scanning electron microscope (SEM) and thermogravimetric analyzer (TGA). In addition, the OTR and WVTR of the illite-PE composite film were 8315 mL/m^2^·day, and 13.271 g/m^2^·day, respectively. The in vitro cytotoxicity of the illite-PE composite film was evaluated using L929 cells, and showed a cell viability of more than 92%. Furthermore, the freshness-maintaining property was tested for a packaging application with bananas; it was found to be about 90%, as indicated by changes in the color of the banana peel, after 12 days.

## 1. Introduction

Recently, freshness-maintaining technologies have been developed to extend the shelf lives of foods or fruits without contamination [1,2,3,4,5,6,7]. In particular, fruits and vegetables produce unusual smells due to breathing when stored in a package, or ripen more quickly due to the release of ethylene gas, resulting in poor marketability [8]. To solve this problem, a laser processing technique was used to create microperforations in polymer packaging films to improve freshness by enhancing the oxygen permeability of fruits and vegetables [9,10]. However, microperforation technology has some disadvantages. It can degrade the film’s tensile strength, making it more prone to tearing and allowing the infiltration of external contaminants such as moisture to occur [11]. As another packaging technique, many inorganic ceramic powders have been used in polymer packaging films to improve their properties such as mechanical strength, gas permeability control, and thermal stability. Porous ceramics such as silica and zeolite have been applied to packaging films to absorb carbon dioxide, ester, and ethylene gas emitted due to the breathing of fruits and vegetables.

This study introduces a method of increasing the oxygen permeability by adding a ceramic filler during the film manufacturing process to generate fine pores at the interface between the film matrix and the filler. Illite materials, which have a plate-like structure, were used for this purpose. According to previous studies, when a plate-like clay mineral is peeled off into nano units and added to a polymer film, the plate-shaped particles with a high aspect ratio complicate the permeation path of gas or water, resulting in decreased oxygen permeability [12,13,14]. However, in this study, we used microsized illite particles with a low aspect ratio to increase oxygen permeability. In addition, when the illite material is mixed with polyethylene (PE) without any pretreatment, the interaction with the polymer is reduced due to the low surface area. Moreover, as agglomeration occurs due to the nature of the ceramic powder, it is not uniformly dispersed in the film, which results in decreased mechanical strength. To address this problem, in this study, a jet mill was used to reduce the particle size of the illite material down to 10 μm or less.

The illite–PE masterbatch was prepared using a twin-screw extruder, and then a illite–PE composite film was manufactured with the blown method [15,16]. The contents and dispersity of illite in the illite–PE masterbatch and film were characterized using a scanning electron microscope (SEM), thermogravimetric analyzer (TGA), and Fourier transform infrared (FT-IR) spectrophotometer. The oxygen and moisture permeability of the illite–PE film and control PE film were determined by assessing the oxygen transmission (OTR) and water vapor transmission rates (WVTR) [17,18,19,20,21]. The in vitro cytotoxicity was evaluated using an MTT assay; L929 mouse fibroblast cells were exposed to various concentrations of the extracts from illite and illite-PE composite film. In addition, the freshness-maintaining property was demonstrated for a packaging application with bananas; the color change of the peel was monitored for 12 days after packing with a control PE film and a manufactured illite-PE composite film.

## 2. Materials and Methods

### 2.1. Chemical and Materials

PE with a density of 0.915 g/cm^3^ used to prepare the masterbatch was purchased from LG Chem. in Seoul, Korea. The melting flow rate (MFR) of the PE measured using ASTM D1238 was 52.0 g/10 min at 190 °C and a load of 2.16 kg. The illite material was obtained from the Yongkoong Illite company in Youngdong, Korea. It was pulverized prior to the production of the masterbatch, and the particle size and specific surface area were confirmed using a particle size analyzer (Horiba, Kyoto, Japan) and Brunauer–Emmett–Teller (Micromeritics, Norcross, GA, USA), respectively.

### 2.2. Preparation of Illite-PE Masterbatch

To control the illite content (30 wt.%) of the masterbatch, the feed rate of the pure low-density PE pellets and the illite required in two feeders (Bautek, Pocheon, Korea) was measured prior to the experiment. To produce the illite-PE masterbatch, illite and PE were introduced into a corotating (Bautek, Pocheon, Korea), twin-screw extruder (Bautek, Pocheon, Korea) at a ratio of 30–70% under a screw speed of 480 rpm at a temperature of 80–190 °C. The extrudate from the screw was pelleted using a pelletizer (Bautek, Pocheon, Korea) (speed 4.3 Hz), yielding the final masterbatch [22,23,24,25].

### 2.3. Preparation of Illite-PE Composite Film

To produce the illite–PE composite film, the illite–PE masterbatch was mixed with PE pellets at a selected ratio using a blown film maker (Hankook E.M., Pyeongtaek, Korea) under screw speeds of 16.8 Hz at a temperature of 130 °C. The final amount of the illite–PE composite film was adjusted to 1 wt.%–2 wt.%. The film thickness was about 0.05–0.06 mm. The surface morphology of the composite film was observed by SEM (Tescan, Brno, Czech).

### 2.4. Assessments of Freshness-Maintaining Performance of Illite-PE Composite Film

Bananas of similar size and ripening stages were selected, packaged, and sealed one-by-one in a control film and an illite-PE composite film, and the changes were observed with the naked eye every four days. Observations continued until the 12th day, and the color differences between the two films were compared by measuring the chromaticity of the banana before and after packing using a colorimeter (Cortex Technology, Hadsund, Denmark). The chromaticity value is L*, a unit of measurement of brightness in the CIE color system, which is generally used in the food industry. The L* value ranged from 0 (black) to 100 (white). The storage conditions were maintained at a temperature of 24 °C and a humidity of 50 % RH [26].

### 2.5. In Vitro Cytotoxicity of Illite-PE Composite Film

To evaluate the biological stability of illite-PE packaging film, a cytotoxicity test was carried out according to an elution test method (ISO 10993-5). The test solution was eagle minimum essential medium (MEM) supplemented with 10% fetal bovine serum (FBS) (Welgene, Gyeongsan, Korea) and 2% penicillin (Gibco, Carlsbad, CA, USA) Sample (illite materials, illite-PE film) were eluted in the test solution at 37 °C for 24 h under continuous stirring. L929 mouse fibroblast cells (ATCC No. CCL-1) were cultured in test solution in a humidified 5% CO_2_ incubator at 37 °C for 24 h. L929 cells were seeded into 96-well plates (10^4^ cells/well) and incubated at 37 °C for 24 h. Then, the cells were treated with extracts of illite materials and the illite-PE composite film at various concentrations (12.5, 25, 50, and 100%) and incubated in a cell incubator at 37 °C. After incubation for 24 h, the cytotoxicity of each extract was measured using a microplate reader (Tecan, Mannedorf, Switzerland) at a wavelength of 570 nm.

### 2.6. Instrumental Analysis

Illite pulverization was conducted at a feed rate of 0.43 kg/h and air pressure of 0.74 Mpa, using a Nisshin SJ-500 (Nisshin, Tokyo, Japan). A particle size analysis was conducted using the wet method of the LA-960 from HORIBA. The specific surface area was measured using a MicroStar^®^ TriStar II 3020 (Micromeritics, Norcross, GA, USA); nitrogen was used for adsorption. For the XRD analysis, Rigaku’s MiniFlex600 (Rigaku, Tokyo, Japan) was used to analyze the diffraction pattern in the range of 5–90° at a rate of 10°/min [27]. A thermogravimetric analysis was performed from 25 to 700 °C at a rate of 10°/min in a nitrogen atmosphere using a Q600 TA instrument (TA instruments, New Castle, DE, USA) [28]. FT-IR analysis was conducted using Jasco’s FT-IR 460 Plus equipment (Jasco, Easton, PA, USA) with the attenuated total reflection method.

An SEM was used to scan the sample surface with a TESCAN MIRA3, which resulted in the dispersibility of the illite contained in the polymer matrix of the masterbatch and film. The masterbatch sample was cut parallel to the ejected direction and the surface was photographed at 5 kV after being platinum-coated. The film sample was cut into a specimen size of 5 mm, and its surface was photographed at 10 kV after platinum-coating the surface in the same manner as that for the masterbatch. The tensile strength of the film was measured under a test speed of 500 mm/min in accordance with the KS M 3001: 2001 standard using a DUT-3000CM (Daekyung, Bucheon, Korea).

To determine the moisture permeability of the film, the amount of moisture that penetrated an area of 33.18 cm^2^ was measured five times under the conditions of 90% RH, 38 °C temperature, and 60 min test interval using a W3/031 (Labthink, Medford, MA, USA), in line with the ASTM E96 standards. To determine the oxygen permeability of the film, the amount of oxygen permeating 1.131 cm^2^ of the specimen for 30 min was measured with a masking technique at 23 °C and 50% RH using a C230 (Labthink, Medford, MA, USA) in accordance with the ASTM D3985 standards. The chromaticity of the banana was measured with a DSM III (Cortex Technology, Hadsund, Denmark) at three particular spots; the measured values at these spots were averaged to produce the chromaticity value of the banana.

## 3. Results

Figure 1 shows the preparation process of the illite-PE masterbatch and film. Before the preparation of the illite-PE masterbatch, the feed rate was set by measuring the weight of the illite and PE exited through the feeder. The screw rate was adjusted to 480 rpm and extruded by mixing the illite and PE through the feeder. The illite and PE were discharged through the feeder at a constant rate and fed into the twin-screw extruder through the hopper. As the temperature of the extruder was higher than the melting point of PE, the extruder was mixed and extruded with the illite while the PE was molten. The extrudate was sufficiently cooled in the water-cooling zone and cut with a pelletizer to obtain a masterbatch. The feed and screw rate of the sample should be set such that the sample does not accumulate in the hopper. The pelletizer rate should be set so that the extrudates are not cut and the proper thickness is maintained. The prepared masterbatch was fed to a blown film maker with LDPE in a constant ratio to prepare an illite-PE composite film.

Figure 2a shows the XRD patterns of illite material, PE, illite-PE masterbatch, and illite-PE composite film. The XRD patterns of the illite material showed peaks at 8.8°, 17.7°, 19.8°, 26.6°, and 45.4°, while those of PE material showed peaks at 19.8°, 21.5°, and 23.9°. The illite-PE masterbatch showed a broad peak at 21.5°, along with the peak of the illite material. In the illite-PE composite film, the illite peak became weak while the film exhibited strong peaks at 21.5° and 23.9°, which were attributable to the increased content of PE. Stronger PE peaks were observed in the illite-PE film (PE content of 98 wt.%) as compared to the illite-PE masterbatch (PE content of 70 wt.%). Figure 2b shows the FT-IR spectrum in the range of 4000–650 ${\mathrm{cm}}^{-1}$ for illite material, PE, illite-PE masterbatch, and illite-PE film. The spectrum of illite material showed a peak of 1000 ${\mathrm{cm}}^{-1}$ (Si-O-Si stretching vibration) and 3600 ${\mathrm{cm}}^{-1}$ (H-O-H stretching vibration). In the spectrum of PE, 2920 ${\mathrm{cm}}^{-1}$, 2850 ${\mathrm{cm}}^{-1}$ (C-H stretching) peaks were observed [29]. In the illite-PE composite film, 1000 ${\mathrm{cm}}^{-1}$, 2900 ${\mathrm{cm}}^{-1}$ and 2850 ${\mathrm{cm}}^{-1}$ peaks appeared. As a result, it was shown that the peak of 1000 ${\mathrm{cm}}^{-1}$ weakens as the illite content in the polymer decreases.

Figure 3 shows the SEM images and TGA analyses of illite, illite-PE masterbatch and illite-PE composite film. The SEM images show the 3 μm of particle size of illite, dispersed in the PE matrix of illite-PE masterbatch, and uniformly dispersed inside the PE matrix without agglomeration in illite-PE composite film [30]. A sudden drop in weight near 400 °C was due to the combustion of PE. The 50% mass loss temperatures of LDPE, illite-PE masterbatch and illite-PE composite film were determined to be 465 °C, 477 °C, and 473 °C, respectively. The illite content in the film was calculated in the section where it was stabilized again after 500 °C. The thermogravimetric analysis showed that the illite content in the masterbatch was about 30%, while that in the composite film was about 1–2%.

Figure 4a shows the oxygen permeability and moisture permeability of the illite-PE composite film by oxygen transmission rate (OTR) and water vapor transmission rate (WVTR) [31]. The illite-PE composite film had a permeability value of 8315 mL/m^2^·day, while the oxygen permeability value of the illite-PE film was about 2 times higher (4090 mL/m^2^·day) [32]. This was attributed to the addition of illite particles between the PE matrices, which ensured a simple path through which oxygen could easily travel through the PE–illite particle interface. The moisture permeability of the illite-PE composite film was determined as the average of the five measured values. The results showed that the moisture permeability was 5.35 g/m^2^·day for control PE and 13.27 g/m^2^·day for the illite-PE composite film. Figure 4b shows the tensile strength of the film: 25.6 Mpa for the illite-PE composite film, which is higher than the value of 17.2 Mpa for the control PE film. This is attributable to the load distribution in the illite due to the interfacial stress between the polymer matrix and the illite particles [33]. In addition, the fact that the tensile strength was increased indirectly indicates that the illite particles were well-dispersed in the polymer matrix.

Figure 5 shows the in vitro cytotoxicity assessment for illite and illite-PE composite film using MTT assay, where L929 mouse fibroblast cells were exposed to the various concentrations of the extracts from illite and illite-PE composite film. They were shown to be capable of metabolizing a dye as 3-(4,5-dimethylthiozol-2-yl)-2,5-diphenyl tetrazolium bromide. Cell viability was calculated using the following Equation (1).
(1)$$\mathrm{Cell}\;\mathrm{viability}\;(\%)=\frac{100\times OD_{570t}}{OD_{570b}}$$
where OD_570t_ and OD_570b_ are the average values of absorbance of the blank solution.

After 24 h of posttreatment, L929 mouse fibroblast cells showed excellent viabilities, i.e., 92.3% and 100%, even up to a concentration of 50% illite and illite-PE composite film, respectively. These results clearly showed the absence of any noticeable toxicity of illite and illite-PE composite film. However, cell viability was decreased to 59% and 93% in the extract solution with a 100% concentration of the illite and illite-PE composite film. This was attributed to the spontaneous MTT reduction which occurred because dehydrogenase enzymes which were only present in viable cells reduced the yellow MTT to purple insoluble formazan crystals, which then accumulated inside the cell [34].

Figure 6 shows the results of the evaluation of the freshness-maintaining function of the illite–PE composite film. The changes in the state of bananas in the control PE film and the illite–PE composite film were checked once every 4 days. The bananas packaged in the control PE film began to change starting on the 4th day, and were severely changed in terms of decay and the color of the peel on the 12th day. The bananas packaged in the illite-PE composite film began to change starting on day 8, and part of them turned black on day 12. The chromaticity of the banana was 46.6 and 46.5 for the control and illite-PE composite film at the beginning, respectively, and 13.0 and 40.7 after 12 days. Thus, the changes were 72.1% and 12.5% over 12 days, respectively. As a result, both visually and in terms of chromaticity, the bananas stored in illite-PE composite film had a lower degree of ripening and decay compared to those stored in the control PE film.

## 4. Conclusions

Ceramic composite films containing PE polymer and illite minerals were prepared to increase the freshness maintenance period of fruits and vegetables inside packaging materials. The illite content in the illite-PE composite films adjusted using the masterbatch was 1.8%, according to TGA analysis. The oxygen permeability value of the illite-PE composite film was 8315 mL/m^2^·day for the illite-PE composite film, which was about 2 times higher than that of an ordinary PE film (4090 mL/m^2^·day). The moisture permeability values of the film were 5.35 g/m^2^·day for ordinary films and 13.27 g/m^2^·day for Illite-PE composite films. The tensile strength values were 25.6 Mpa for the illite-PE composite film, which was higher than those of the control film (17.2 Mpa). In the storage experiment of the banana, the ripening difference between the control film and illite-PE composite film was observed from the 4th day of the experiment. On the 12th day, there was a noticeable decrease in the extent of the decay and color change of the bananas packaged in the illite-PE composite film compared to those packaged in the control film. In view of these results, because the oxygen permeability increased when the composite film containing illite was used as a packaging material, it is expected to reduce the oxygen concentration inside the package by breathing or increase the shelf life of vegetables and fruits that release aging-expediting gases.

## Figures and Tables

**Figure 1 materials-13-02382-f001:**
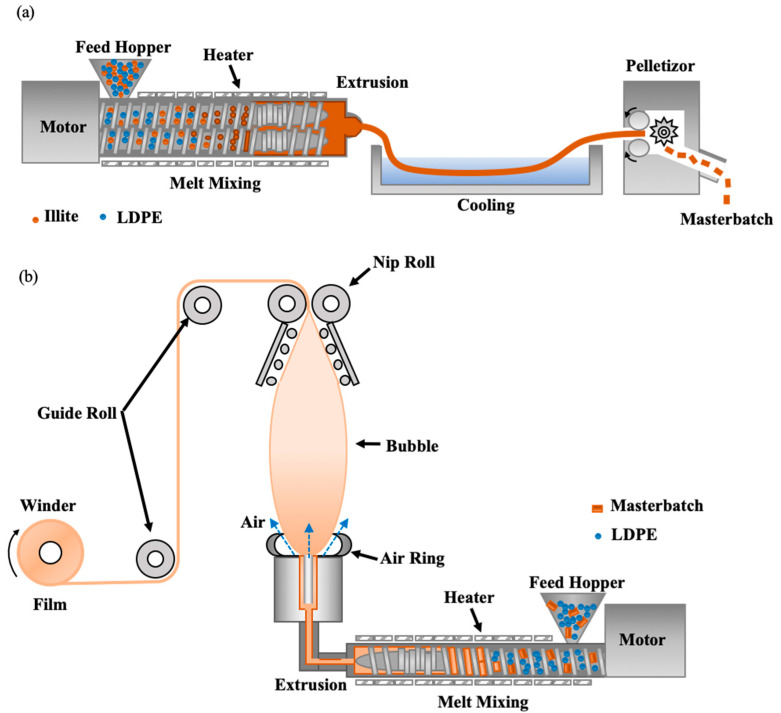
Schematic diagrams for preparation of (**a**) illite-PE masterbatch and (**b**) illite-PE film.

**Figure 2 materials-13-02382-f002:**
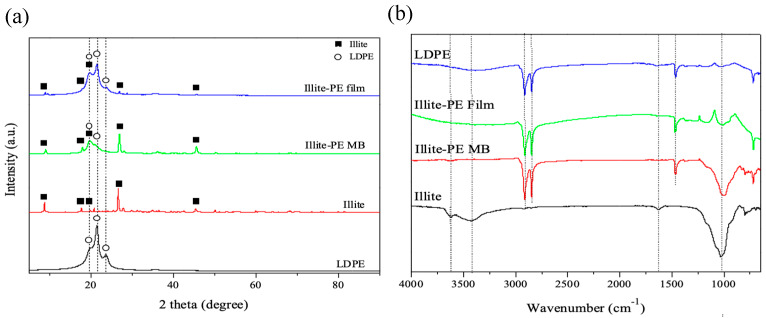
(**a**) XRD and (**b**) FT-IR spectra of illite, iliite-PE masterbatch, and iliite-PE film.

**Figure 3 materials-13-02382-f003:**
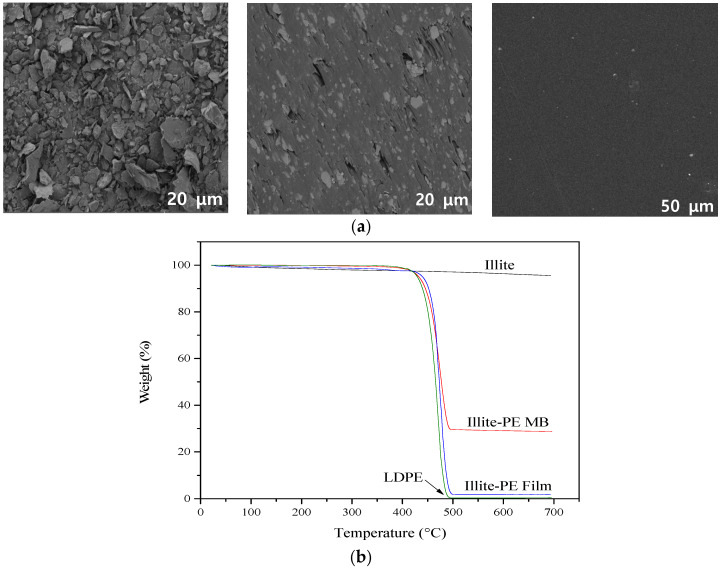
(**a**) SEM images and (**b**) TGA analysis of illite, iliite-PE masterbatch, and iliite-PE film respectively.

**Figure 4 materials-13-02382-f004:**
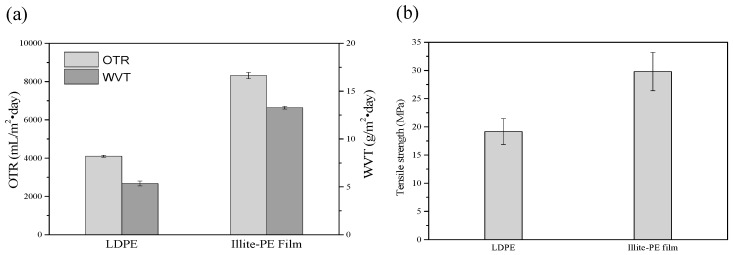
Comparison of (**a**) oxygen transmission rate (OTR) and water vapor transmission rate. (WVTR), and (**b**) tensile strength of control LDPE and illite-PE film.

**Figure 5 materials-13-02382-f005:**
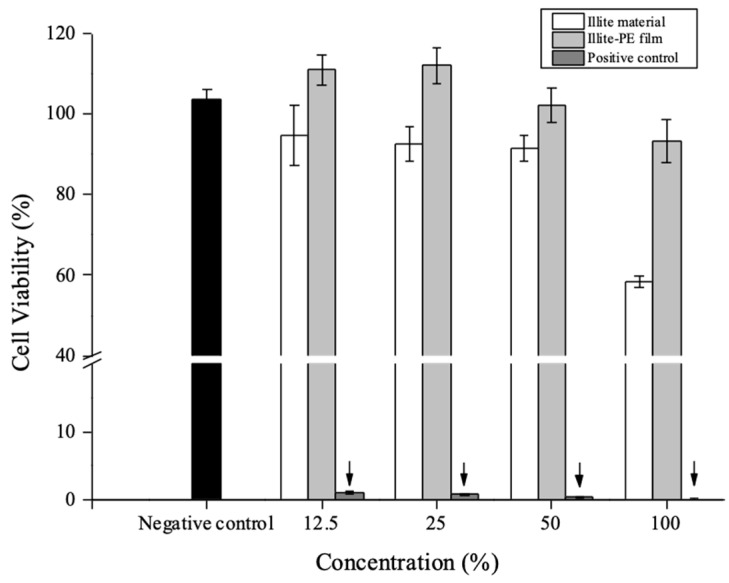
In vitro cytotoxicity of illite material and illite-PE film (negative control; high density. polyethylene film, positive control; 0.1 % ZDEC polyurethane film).

**Figure 6 materials-13-02382-f006:**
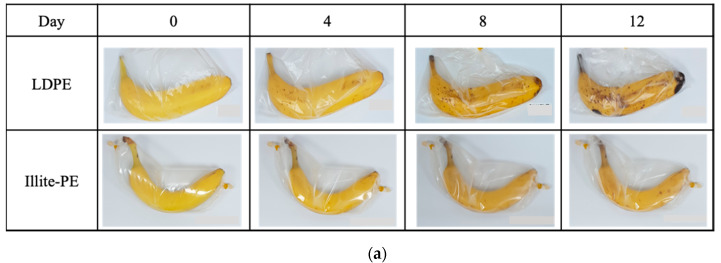
Comparison of (**a**) freshness-maintenance property and (**b**) chromaticity changes of illite–PE film and control LDPE for 12 days.

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
