# Peer review of "High Oxygen and Water-Vapor Transmission Rate and In Vitro Cytotoxicity Assessment with Illite-Polyethylene Packaging Films"

_materials, 2020, doi:10.3390/ma13102382_

Round 1
Reviewer 1 Report
This work is well organized and in good level presented. I suggest publication after the following revisions.
XRD results,lines:154-156 The authors write: "In the illite-PE
composite film, the illite peak became weak and the film exhibited strong peaks at 21.5° and 23.9°, which are attributable to the increased content of PE"
As I can see no peak shift for illite's peak's is observed. Please add a comment in the text about this. What kind of nanocomposite structure this no peak shift suggest?
TG results: The LDPE TG curve must be added in the diagramm. Temperature of 50% mass loss mast be calculated and added in the reults comments
Reviewer 2 Report
The paper of Seong et al. is well-written and presents interesting results on the possiblity to keep fresh friuts and vegetables much longer, which is which is an extremely important issue with so much food being thrown away. However, still I have few comments on the paper that generally deal with the "quality of the presentation":
- Page 1, line 28,- two words are written with capital letter. Please correct this.
- Page 2, lines 61-63: Something is wrong with this sentnce: "..with bananas and observed the color change of the peel was observed". Maybe this should be rephrased.
- Point 2.1. Line 69. Please re-write the sentence to "The melt index and density were 24.0 g/10 min (190, 2.16 kg) and 0.915 g/cm3, respectively"
- Point 2.2. What was the yield of this step.
- Point 2.3. How did you prepare the composite film sample?
- Point 2.6. Should be MPa instead of Mpa.
- Please unify the form of presenting the numbers in FTIR, as once you write: 2,920, once 2850 etc.
- In Fig. 4 -please provide the "stsitics" of this measurement, by means of the standard deviation. It is crucial to presnt the date in the bar chart with the measurements' error.
- Please unify in the whole paper, if there is a space or there is no space between the number and the unit.
- Why authors choose 12 days as the maximum time of the experiment? Is it according to standard? Do you plan to expand this term, to maybe to shorten it, since the plot with two points (Fig. 6) doesn't make sense. Maybe more mearumenets point will provide more data.
Reviewer 3 Report
I am in favor of publishing this well written manuscript. My comments are given below for a major revision:
- Title: “polyethylene” should be included. I suggest “…Polyethylene/Ceramic Composite Packaging Films”
- Introduction: Rewriting is necessary. Please include more literature studies, for example, regarding modification of PE film for similar applications using different additives and so on. Also, last paragraph is too big detailing current study which should not be a part of this section. Make it concise and clearly mention what is novel in this study.
- Correction: “The melt index and density were 24.0 g/10 min (190, 2.16 kg) and 0.915 g/cm3, respectively.”
- “The illite material was obtained from company in Korea” – which company?
- Section 2.2: Was any of the ingredients dried before extrusion? What was the moisture content of the raw materials?
- Section 2.3: Please list the parameters used here for mixing the melt.
- “The final content of illite in the prepared composite film was adjusted to 1%–2%.” – Did the author optimize this loading? What was the basis?
- Correction: “The surface morphology of the composite film sample was analyzed using a SEM.”
- “Bananas of similar particle size” – Here, “particle” is confusing. Do you mean similar specimen size?
- “The film sample was cut into a particle size of 5 mm” – Similarly, particle size should be replaced by specimen size.
- “The extrudate was sufficiently cooled in the cooling zone” – Please mention what type of material was used as coolant.
- “In the illite-PE composite film, weak 1000 cm-1 peak and strong 2,900 cm-1, 2,850 cm-1 peaks appeared.” – What do you mean by weak and strong peaks? Were those normalized? Please clarify.
- Fig. 2(b): Please use vertical lines to show the peak change/shift.
- “…uniformly dispersed inside the PE matrix without agglomeration…”- SEM images mostly helpful to characterize the local dispersion/ aggregation of the nanoparticles. On the other hand, optical microscopy can detect the dispersion more globally. As a result, OM could be more useful in this case and claiming “uniform” dispersion could be misleading. How many SEM images were taken at different regions of the sample for verification? Please comment on this in the text.
- “Thermogravimetric analysis of the masterbatch and the composite film, where…” – Please include thermogram of PE in fig. 3 (b).
- Fig.4: Please include standard deviation.
- “Fig. 4(b) shows the tensile strength of the film …” – Please include % elongation of break data. For a film application, %Eb is also an important property to consider.
- The English language of this manuscript needs improvement. Some examples are given above but there are more in the manuscript. I am recommending a proofread of the manuscript thoroughly, or to use a language editing service.
Round 2
Reviewer 3 Report
Acceptance is recommended.